# Umbilical Artery Thrombosis Masquerading as Single Umbilical Artery in a Stillbirth

**DOI:** 10.3390/diagnostics15010094

**Published:** 2025-01-03

**Authors:** Yin Ping Wong, Rahana Abd Rahman, Ay Eeng Tan, Geok Chin Tan

**Affiliations:** 1Department of Pathology, Faculty of Medicine, Universiti Kebangsaan Malaysia, Kuala Lumpur 56000, Malaysia; 2Department of Obstetrics and Gynaecology, Faculty of Medicine, Universiti Kebangsaan Malaysia, Kuala Lumpur 56000, Malaysia; drrahana@ppukm.ukm.edu.my; 3Obstetrics and Gynaecology, Prince Court Medical Centre, Kuala Lumpur 50450, Malaysia; ayeeng.tan@princecourt.com

**Keywords:** placenta, single umbilical artery, stillbirth, umbilical cord, umbilical artery thrombosis

## Abstract

**Background:** Umbilical artery thrombosis (UAT) masquerading as a single umbilical artery (SUA) is a rare but critical diagnostic challenge in prenatal care. **Case Presentation:** We described a case of a 22-year-old primigravida with an uneventful obstetric history who presented with reduced fetal movements at 22 weeks of gestation. Ultrasound showed no gross fetal structural anomalies while umbilical artery Doppler flow imaging revealed an isolated SUA. The patient again presented with diminished fetal movement at 24 weeks gestation, and a diagnosis of intrauterine demise was confirmed ultrasonographically. She was then induced and delivered a macerated stillborn female fetus. Placental examination revealed three umbilical vessels with an occlusive thrombus seen within the umbilical artery consistent with UAT, a finding previously mistaken for SUA. **Conclusions:** This case underscores the diagnostic difficulties of UAT radiologically, especially when there was no prior documented evidence of two umbilical arteries. Identification of at-risk fetuses would allow for close monitoring or effective interventions to be implemented as early as possible to avert preventable fetal loss.

## 1. Introduction

Prenatal ultrasound assessment of umbilical cord vessels is crucial to identify a missing umbilical artery. Single umbilical artery (SUA), a two-vessel cord, is widely recognised as a prenatal marker for fetal structural malformations, chromosomal abnormalities, fetal growth restriction, and perinatal death [1,2]; it occurs in around 0.2–0.5% of pregnancies. SUA is a result of primary agenesis or secondary atresia or atrophy of one of the previously normal umbilical arteries. SUA is usually diagnosed in the second trimester, which warrants a detailed anatomical ultrasound examination for other anomalies. Isolated SUA is not associated with an increased risk for chromosomal anomaly but vice versa applies if there is the presence of other structural abnormalities [3]. Accumulating evidence has shown that isolated SUA is associated with an increased rate of emergency caesarean delivery for fetal distress, preterm delivery, low birth weight, and being small for the gestational age [4], although these associations were not proven to be significant statistically [5].

Umbilical artery thrombosis (UAT) is an exceedingly rare but serious complication of pregnancy associated with poor fetal outcomes. It has an estimated incidence ranging from 0.0025% to 0.045% of deliveries, reaching up to 0.4% in high-risk pregnancies [6]. To the best of our knowledge, there have only been 282 cases with UAT published in the English literature hitherto. UAT is characterised by thrombotic occlusion within the umbilical artery. Umbilical arterial blood flow obstruction by thrombus disrupts blood flow to the fetus, potentially compromising fetal oxygenation and nutrient delivery. The resultant vascular compromise can manifest with a spectrum of clinical presentations, ranging from subtle abnormalities on prenatal imaging to fetal distress, reduced fetal movement, fetal growth restriction and in severe cases, intrauterine death [6,7]. It can be easily mistaken for an isolated SUA ultrasonographically, especially when there is complete occlusion of the involved vessel, which results in a significant reduction in arterial blood flow [8]. This could represent a potential diagnostic pitfall, particularly when there is no prior documented evidence of two umbilical arteries.

The distinction between UAT and SUA, although challenging, is critical as it carries significant implications in fetal management. Unlike SUA where there is no active intervention other than close follow-up and timely delivery, active intervention such as low-molecular-weight heparin may be administered to the mother with suspected UAT to mitigate the condition [9]. In addition, predisposing factors for the development of UAT can sometimes be identified and treated accordingly to prevent recurrence in the next pregnancy. We herein highlight the diagnostic challenges and clinicopathological features in a rare case of UAT masquerading as SUA that resulted in stillbirth.

A literature search on the PubMed database was conducted using the keywords (umbilical artery thrombosis) AND (fetal outcomes), which yielded 124 articles. Of these, 38 articles (with a total of 282 cases with histologically confirmed UAT) [6,7,8,9,10,11,12,13,14,15,16,17,18,19,20,21,22,23,24,25,26,27,28,29,30,31,32,33,34,35,36,37,38,39,40,41,42,43] were selected and included for review. The clinicopathological characteristics, histopathological findings, and fetal outcomes of these cases are summarised in Appendix A.

## 2. Case Report

A 22-year-old primigravida presented with decreased fetal movement that gradually disappeared at the 24-week period of gestation. A diagnosis of intrauterine demise was made by ultrasonography, confirming the absence of fetal cardiac activity. The current pregnancy was conceived spontaneously. She was not known to have any underlying medical condition. Neither her husband nor the patient herself is a smoker or a drug abuser. This pregnancy had been uneventful until 22 weeks of gestation when fetal growth restriction was detected following a detailed scan with an estimated fetal weight below the 10th percentile for the gestational age. Only one umbilical artery blood flow signal was detected with umbilical artery Doppler flow imaging. The fetus was suspected of having an SUA. Given the association of SUA with concurrent congenital anomalies, a detailed fetal anatomical survey—particularly focusing on the cardiovascular and genitourinary system, fetal echocardiography, and fetal genetic testing—was carried out. However, no fetal structural or chromosomal abnormalities were detected. A diagnosis of isolated SUA was rendered.

She was then induced and delivered a macerated stillborn female fetus with no visible gross congenital anomalies. Postmortem examination was not performed as the parents declined. The entire placenta and umbilical cord (UC) were sent for histopathological evaluation. The placental trimmed weight was 239.3 g, which was large (more than the 90th percentile) for the gestational age (the mean weight for 24 weeks is 189 g; 90%CI: 145 g, 233 g). The umbilical cord was paracentrally inserted, with a total length of 34 cm. The umbilical cord appeared hypercoiled with a coiling index of 0.8 (Figure 1). Intriguingly, the umbilical cord contained three vessels (two arteries and one vein), in contrast to the ultrasound findings. The membrane was complete and appeared unremarkable. Histopathological examination of the placenta revealed immature chorionic villi appropriate for the gestational age. There was no significant inflammation in the fetal membrane or the fetal vessels. Areas with avascular villi were also seen, with associated stem vessel obliteration. One of the umbilical arteries showed complete occlusion by a thrombus (Figure 2). A diagnosis of segmental complete fetal vascular malperfusion (FVM) secondary to UAT as a plausible explanation for the fetal demise was made. The mother, however, defaulted subsequent follow-up and was not screened with the full panel of maternal thrombophilia tests.

## 3. Discussion

Ultrasound examination is the key for the prenatal detection of UAT. A two-dimensional ultrasound transverse section of a normal umbilical cord would reveal three ring structures: one big ring, the umbilical vein, and two small rings, the umbilical arteries. The number of umbilical arteries can be readily determined by blood flow evaluation using colour Doppler at the level of fetal bladder or in an axial view of the umbilical cord [44]. Nonetheless, UAT is diagnostically challenging prenatally, with approximately 25% (68/282) of UAT diagnosed only postnatally via histopathological examination of the umbilical cord [6,7,8,9,10,11,12,13,14,15,16,17,18,19,20,21,22,23,24,25,26,27,28,29,30,31,32,33,34,35,36,37,38,39,40,41,42,43]. Additionally, it is easily misdiagnosed as SUA ultrasonographically when one of the umbilical arteries is obliterated by a thrombus resulting in the absence of umbilical artery blood flow signals. UAT should be suspected when there is only one umbilical artery being detected when there have previously been two [8]. Attending obstetricians should maintain a high index of suspicion of UAT in cases where antenatal ultrasound shows SUA on the colour Doppler imaging but a ”Mickey Mouse” appearance on the transverse section of the cord is demonstrated [45].

The occluded thrombotic artery in parallel with the remaining artery are surrounded by the uterine vein, resembling that of “an orange grabbed by a hand”—the so-called “orange grabbed sign”—an ultrasound finding classically described in UAT [18]. On the contrary, an increasing diameter of the umbilical artery with no significant change in the diameter of the umbilical vein is a characteristic prenatal ultrasound finding of SUA [46]. In suspected cases of UAT, assessment of fetal growth by ultrasound is recommended as UAT is frequently associated with fetal growth restriction.

While different anatomical abnormalities of the umbilical cord, along with maternal or fetal pathologies, are recognised as risk factors attributed to UAT, the exact aetiology of UAT remains unclear in many instances. Fetal vascular thrombus formation may be explained by various combinations of three risk factors (Virchow’s triad)—blood stasis, hypercoagulability, and endothelial wall injury. Endothelial wall damage may be caused by maternal smoking, fetal inflammatory response to infection (funisitis), or meconium-induced vascular necrosis, while hypercoagulability may be associated with acquired or hereditary maternal or fetal thrombophilia, including antiphospholipid antibody syndrome, protein C or S deficiencies, or factor V Leiden mutation [22,37]. Li et al. (2023) suggested that abnormal maternal auto-antibodies may contribute to UAT by causing vascular wall damage, which promotes thrombus formation [33].

Most cases of UAT are related to UC abnormalities, including excessive length, hypercoiling, true knots, cord strictures, abnormal cord insertion, and compression, leading to obstructed umbilical blood flow and the occurrence of UAT [6,9,10,11,13,14,17,18,19,20,21,22,23,27,28,29,30,35,36,37,38,47,48]. The absence of Hyrtl’s anatomosis, a vascular connection between the two umbilical arteries near the placental end of the umbilical cord insertion, was believed to aggravate the effects of UAT. The absence of Hyrtl’s anastomosis would lead to a marked reduction in fetal blood flow in cases with single UAT, resulting in severe fetal hypoxia and potentially fatal outcomes [6,27]. Other obstetric risk factors that are potentially associated with UAT are gestational diabetes mellitus, hypertensive disorders of pregnancies, and Rhesus alloimmunisation [26].

Histological examination of the umbilical cord following delivery is the gold standard for the diagnosis of UAT, demonstrating the presence of a thrombus within the umbilical artery. In addition, total or partial necrosis of the affected umbilical arterial wall can be seen, with myocytes hypereosinophilia, pyknotic nuclei, nuclear fragmentation, and loss of nuclear details [28], while the remaining patent umbilical artery may show oedema of the tunica intima, an indirect sign of hypoxia [18].

Sections from the placenta often reveal evidence of impaired fetal perfusion, which includes stem villi vascular thrombosis, avascular villi, villous–stromal karyorrhexis (VSK), and increased circulating nucleated fetal red blood cell precursors and are collectively termed FVM [48,49]. FVM can be separated into two distinct patterns: global/partial and segmental/complete. Stasis, due to umbilical cord obstruction, is the most common underlying risk factor for both patterns. Unlike involutional fetal vascular changes secondary to stillbirth, preexisting FVM in a stillbirth shows temporal–spatial heterogeneity of the distal villous lesions (avascular villi and/or VSK), sharply divergent from the surrounding villi in their stage of involution. VSK, an initial phase of ischaemic injury, is characterised by villous endothelial karyorrhexis and congestion, stromal apoptosis, fragmentation, and extravasation of red blood cells into the villous stroma; it will then progress towards hyalinised avascular villi at a later stage [48]. Besides FVM, other histopathological findings such as villitis, chorionic villous hypervascularity, and subchorionic villous infarction were previously described in placentas complicated by UAT [7].

FVM with an associated compromised umbilical cord has an overall low recurrence risk. Although umbilical cord abnormalities are generally thought to be a sporadic event, rare examples of recurrent intrauterine fetal death related to hypercoiled and excessive long cords have been reported, suggesting the possibility of an underlying genetic etiology in a subset of cases [50]. The risk of recurrence of UAT is modestly increased in cases with predisposing maternal or fetal coagulopathies such as antiphospholipid antibody syndrome, protein C or S deficiencies, or factor V Leiden mutation [48]. In pregnancies previously complicated with cord pathologies such as hypercoiling, abnormal cord insertion, or placental FVM, on top of standard clinical Doppler ultrasound parameters of the umbilical arteries, recent studies have suggested using the measurement of the wave reflections in the umbilical artery (umbilical artery pulsatility index) using Doppler ultrasound to predict placental vascular pathology antenatally, including FVM, an important risk of stillbirth that requires immediate clinical attention and close monitoring [51].

There is no consensus with regard to the management of UAT. There is no cure for UAT other than delivery, which in most cases reported in the literature is typically performed via emergency caesarean section. The choice between an urgent delivery or expectant management largely depends on the gestational age, the cord condition, the fetus’ well-being, as well as the experience and understanding of the attending obstetricians. If gestational age is favourable, prompt delivery with antenatal corticosteroid administration (if preterm) is recommended following the ultrasound diagnosis of UAT. In cases where the fetal condition is reassuring, the option to prolong pregnancy to the third trimester under strict fetal monitoring could be contemplated [7]. In cases where there are obvious cord abnormalities, more aggressive management should be offered [31]. The couple should also be counselled with regards to the risks and benefits of an urgent premature delivery versus a carefully monitored watch-and-wait approach. Alternatively, a possible therapeutic role of low-molecular-weight heparin until delivery has recently been documented [9], however, this still requires future studies for validation.

## 4. Conclusions

In summary, this case highlights the diagnostic difficulties of UAT radiologically, especially when there is no prior documented evidence of two umbilical arteries. Further studies are warranted to elucidate the pathophysiological mechanisms of UAT, refine its diagnostic strategies, and enhance therapeutic interventions towards improving perinatal outcomes in similar clinical scenarios. Identification of at-risk fetuses with UAT would allow for close monitoring or effective timely therapeutic interventions to be implemented to avert preventable fetal loss.

## Figures and Tables

**Figure 1 diagnostics-15-00094-f001:**
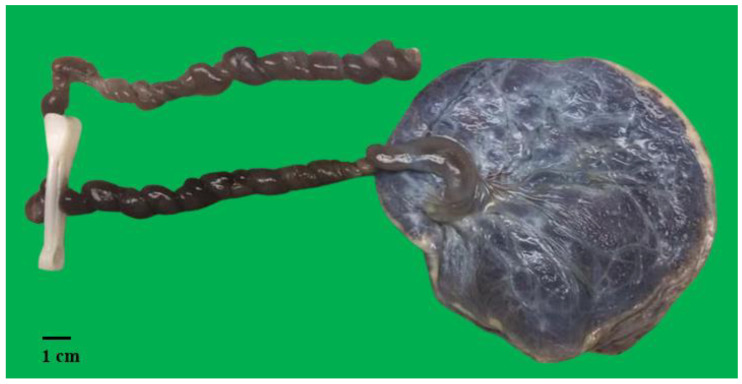
Gross placental examination of a stillbirth demonstrating hypercoiled umbilical cord, with a coiling index of 0.8.

**Figure 2 diagnostics-15-00094-f002:**
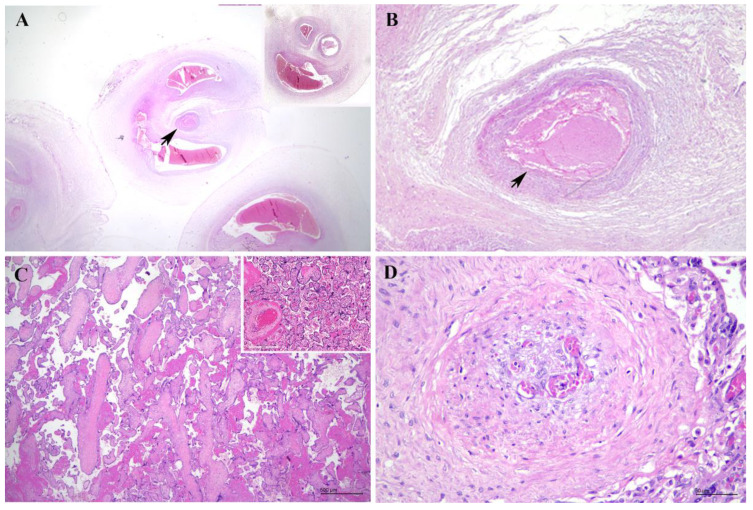
Placental examination of a stillbirth complicated with umbilical artery thrombosis. (**A**) A section of umbilical artery showing near complete occlusion by thrombus (black arrow) (Haematoxylin and Eosin (H&E), ×40), inset shows normal umbilical cord with patent vessels for comparison. (**B**) Intraluminal thrombus (black arrow) at higher magnification (H&E, ×200). (**C**) A section from full thickness of placenta revealed avascular villi (H&E, ×100), inset shows normal placenta comprising chorionic villi with patent vasculature for comparison; and (**D**) associated stem vessel obliteration (H&E, ×600).

## Data Availability

The original contributions presented in the study are included in the article/Appendix A, further inquiries can be directed to the corresponding author.

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
