# Peer review of "Umbilical Artery Thrombosis Masquerading as Single Umbilical Artery in a Stillbirth"

_diagnostics, 2025, doi:10.3390/diagnostics15010094_

Round 1
Reviewer 1 Report
Comments and Suggestions for Authors
Dear Editor thank you for giving me this chance to review this study entitles “ Umbilical Artery Thrombosis Masquerading as Single Umbilical Artery in a Stillbirth”
Here are my comments:
Lines 66 and 67 it was written: “had an unremarkable pregnancy course until current presentation.” What does mean by that? But it 22 weeks IUGR had detected.
What was the smoking and drug use history of mother? Or her husband?
Please in figure 2 by arrow demonstrate “umbilical artery thrombosis”
What is the “H&E” please before use of that add “hematoxylin and eosin (H&E)”.
Given that the mother was without any risk factor, what is the message of this study to clinicians? What criteria clinicians should pay attention to? it is needed to authors give a clue for the prevention of this problem. According to the authors, what was the main cause of this problem?
Line 186: who are the “at-risk fetuses’
How many cases have been reported until now?
discussion needs to be re-written.
Author Response
Comment 1: Lines 66 and 67 it was written: “had an unremarkable pregnancy course until current presentation.” What does mean by that? But it 22 weeks IUGR had detected.
Our response: Thank you for the query. The current pregnancy had been uneventful until 22 weeks when FGR was detected ultrasonographically. We’d reworded and rearranged the sentences for clarity. See page 2, lines 78 – 81.
Comment 2: What was the smoking and drug use history of mother? Or her husband?
Our response: Thank you for the questions. As far as we knew, neither the patient nor her husband smoke and are drug abusers. We’d added in a statement in the case history section for clarity. See page 2, lines 77 – 78.
Comment 3: Please in figure 2 by arrow demonstrate “umbilical artery thrombosis”
Our response: Thank you for the suggestion. We’d added in an arrow demonstrating the thrombosis in figure 2. Figure legend is also updated accordingly. See page 3, Figure 2.
Comment 4: What is the “H&E” please before use of that add “hematoxylin and eosin (H&E)”.
Our response: Yes, H&E means haematoxylin and eosin. It was corrected as suggested. See page 3, line 115.
Comment 5: Given that the mother was without any risk factor, what is the message of this study to clinicians? What criteria clinicians should pay attention to? it is needed to authors give a clue for the prevention of this problem. According to the authors, what was the main cause of this problem?
Our response: Thank you for the suggestion. Although different anatomical conditions of the umbilical cord and maternal or fetal pathologies are considered risk factors, the etiology of umbilical cord thrombosis remains unclear in many cases. To our opinion, the cause of stillbirth in this case was cord-related, as demonstrated by the presence of hypercoiled cord. A suggestion to clinicians attending pregnancies previously complicated with cord pathologies is added in the discussion section. Please see page 4, lines 144 – 146 and page 5, lines 196 - 202.
Comment 6: Line 186: who are the “at-risk fetuses’
Our response: “at-risk fetuses” is referring to “fetuses with UAT”, and was added accordingly. Please see page 6, line 227.
Comment 7: How many cases have been reported until now?
discussion needs to be re-written.
Our response: Thank you for the suggestion. We’d added in a literature review of all cases of UAT that had been reported so far. A few sentences and a supplementary Table and new references were added for this purpose. Please see page 1, 2, lines 45 – 46, lines 66 – 70; page 4, lines 126 – 128, lines 152 – 154, lines 159 – 163; page 5, lines 204 – 206, lines 214 – 215; supplementary Table S1 and references.
Reviewer 2 Report
Comments and Suggestions for Authors
This is a good case description of a well-known topic and event. The authors should concentrate more on differential diagnosis of stillbirth rather than that of single umbilical artery. Did they do uterine artery Doppler in this case? If data are available please add this. There is a major recent paper showing that uterine arteries Doppler provides insight on the etiology of stillbirth and can anticipate histological anomalies on the placenta. Please add a sentence along with this important concept and reference (ref 1).
Was umbilical artery spectral Doppler normal?
Please Doppler requires capital letters.
For the rest I am happy with the paper.
Reference
1. Amodeo S, Cavoretto PI, Seidenari A, Paci G, Germano C, Monari F, Donno V, Giambanco L, Avagliano L, Di Martino D, Fusé F, Masturzo B, Chiantera V, Facchinetti F, Ferrazzi E, Candiani M, Bulfamante G, Farina A. Second trimester uterine arteries pulsatility index is a function of placental pathology and provides insights on stillbirth aetiology: A multicenter matched case-control study. Placenta. 2022 Apr;121:7-13. doi: 10.1016/j.placenta.2022.02.021. Epub 2022 Feb 26. PMID: 35245721.
Comments on the Quality of English LanguageAdequate
Author Response
Comment 1: This is a good case description of a well-known topic and event. The authors should concentrate more on differential diagnosis of stillbirth rather than that of single umbilical artery. Did they do uterine artery Doppler in this case? If data are available please add this. There is a major recent paper showing that uterine arteries Doppler provides insight on the etiology of stillbirth and can anticipate histological anomalies on the placenta. Please add a sentence along with this important concept and reference (ref 1).
Our response: Thank you for your comments. We agree with the reviewer that there are a myriad of causes of stillbirth which include maternal, fetal and umbilical cord causes.
In ref 1 suggested by the reviewer, the paper revealed that uterine arteries Doppler could provide insight on the etiology of stillbirth and can predict histological changes in the placenta specifically referring to maternal vascular malperfusion. Maternal vascular malperfusion, a histological change seen in the placenta frequently related to maternal causes of stillbirth such as uncontrolled maternal hypertension, pre-eclampsia, etc. Nonetheless, in this paper, the cause of stillbirth was obviously cord-related, leading to fetal vascular malperfusion changes in the placenta (please see page 3, lines 104 – 105). We felt that discussion along with maternal vascular malperfusion as a cause of stillbirth is out of scope for this case report.
We’d however included another more suitable reference and a sentence in the discussion to reflect to readers the importance of uterine arteries Doppler in predicting cord pathology-related fetal vascular malperfusion changes in the placenta - which is more relevant to the present manuscript. Please see page 5, lines 196 – 202.
Comment 2: Was umbilical artery spectral Doppler normal?
Our response: Thank you for the question. Spectral Doppler unfortunately was not done for this patient.
Comment 3: Please Doppler requires capital letters.
Our response: Thank you. We’d changed it accordingly. Please see page 4, line 133.
For the rest I am happy with the paper.
Reference
- Amodeo S, Cavoretto PI, Seidenari A, Paci G, Germano C, Monari F, Donno V, Giambanco L, Avagliano L, Di Martino D, Fusé F, Masturzo B, Chiantera V, Facchinetti F, Ferrazzi E, Candiani M, Bulfamante G, Farina A. Second trimester uterine arteries pulsatility index is a function of placental pathology and provides insights on stillbirth aetiology: A multicenter matched case-control study. Placenta. 2022 Apr;121:7-13. doi: 10.1016/j.placenta.2022.02.021. Epub 2022 Feb 26. PMID: 35245721.
Reviewer 3 Report
Comments and Suggestions for Authors
This is an interesting case report. I would suggest the following:
1. Provide an image of the umbilical cord obtained with diagnostic ultrasound so the reader can determine if here are any "hints" of thrombosis.
2. Since the authors most likely have access to normal cord and placental images of the microscopic images (Table 2) I would suggest they provide a side-by-side image of normal and pathology which are labeled. For the reader who is not a pathologist, this would help to illustrate the differences between normal and abnormal for Figure 2.
Author Response
This is an interesting case report. I would suggest the following:
Comment 1: Provide an image of the umbilical cord obtained with diagnostic ultrasound so the reader can determine if here are any "hints" of thrombosis.
Our response: Thank you for your suggestion. We would like to comply to the reviewer’s suggestion to provide an image of the ultrasound findings on the cord. Unfortunately the image of the cord was not taken during the patient's presentation, and is not available for this purpose.
Comment 2: Since the authors most likely have access to normal cord and placental images of the microscopic images (Table 2) I would suggest they provide a side-by-side image of normal and pathology which are labeled. For the reader who is not a pathologist, this would help to illustrate the differences between normal and abnormal for Figure 2.
Our response: Thank you for your comments. Yes, we do have access to normal cord and placental microscopic images in our centre. Figure 2 was changed with additional inset as suggested. Figure legend is also updated accordingly. See page 3, Figure 2.